# A Critical Role of Intracellular PD-L1 in Promoting Ovarian Cancer Progression

**DOI:** 10.3390/cells14040314

**Published:** 2025-02-19

**Authors:** Rui Huang, Brad Nakamura, Rosemary Senguttuvan, Yi-Jia Li, Antons Martincuks, Rania Bakkar, Mihae Song, David K. Ann, Lorna Rodriguez-Rodriguez, Hua Yu

**Affiliations:** 1Department of Immuno-Oncology, Beckman Research Institute, City of Hope National Medical Center, Duarte, CA 91010, USA; ruhuang@coh.org (R.H.); yili@coh.org (Y.-J.L.); amartincuks@coh.org (A.M.); 2Department of Surgery, City of Hope National Medical Center, Duarte, CA 91010, USA; bnakamura@coh.org (B.N.); rsenguttuvan@coh.org (R.S.); misong@coh.org (M.S.); 3Department of Pathology, City of Hope National Medical Center, Duarte, CA 91010, USA; rbakkar@coh.org; 4Department of Diabetes Complication and Metabolism, Arthur Riggs Diabetes & Metabolism Research Institute, City of Hope National Medical Center, Duarte, CA 91010, USA; dann@coh.org

**Keywords:** intracellular PD-L1, ovarian cancer, progression

## Abstract

Disrupting the interaction between tumor-cell surface PD-L1 and T cell membrane PD-1 can elicit durable clinical responses. However, only about 10% of ovarian cancer patients respond to PD-1/PD-L1 blockade. Here, we show that PD-L1 expression in ovarian cancer-patient tumors is predominantly intracellular. Notably, PARP inhibitor treatment highly increased intracellular PD-L1 accumulation in both ovarian cancer-patient tumor samples and cell lines. We investigated whether intracellular PD-L1 might play a critical role in ovarian cancer progression. Mutating the PD-L1 acetylation site in PEO1 and ID8*^Brca1−/−^* ovarian cancer cells significantly decreased PD-L1 levels and impaired colony formation, which was accompanied by cell cycle G2/M arrest and apoptosis induction. PEO1 and ID8*^Brca1−/−^* tumors with PD-L1 acetylation site mutation also exhibited significantly reduced growth in mice. Furthermore, targeting intracellular PD-L1 with a cell-penetrating antibody effectively decreased ovarian tumor-cell intracellular PD-L1 level and induced tumor-cell growth arrest and apoptosis, as well as enhanced DNA damage and STING activation, both in vitro and in vivo. In conclusion, we have shown the critical role of intracellular PD-L1 in ovarian cancer progression.

## 1. Introduction

Ovarian cancer is one of the most lethal gynecological malignancies due to its poor prognosis, with a 5-year survival rate of less than 50% and a 10-year survival rate of approximately 35% [1]. In addition to standard surgical debulking and adjuvant chemotherapy, targeted therapies such as poly ADP–ribose polymerase (PARP) inhibitors (PARPi) and anti-VEGF/VEGF inhibitors have significantly reformed the treatment paradigm of ovarian cancer and markedly prolonged patient progression-free survival (PFS) [2,3]. However, recurrence and acquired resistance to PARPi and other therapies remain major challenges. Targeting the interactions between PD-L1 on the tumor-cell surface and PD-1 on T cells with antibodies has shown great promise for significantly prolonging patient survival in several solid tumors, such as advanced metastatic melanoma, non-small cell lung cancer, and renal cell carcinoma [2]. However, the results from multiple clinical trials suggest that targeting PD-L1/PD-1 with current antibodies does not significantly improve the survival of ovarian cancer patients [2]. The overall objective response rate (ORR) to single-agent PD-L1/PD-1 antibody treatment ranges from only 8% to 15% [4,5,6,7].

Currently, the evaluation of PD-L1 expression is based solely on its cell surface level, which serves as a predictor of the PD-L1/PD-1 blockade response [8,9]. Based on its cell-surface expression, ovarian cancer is considered to have low levels of PD-L1 [10]. Intriguingly, two meta-analyses did not reveal any significant associations between the PD-L1 surface expression level and overall survival (OS) or PFS in patients with ovarian cancer. In contrast, PD-L1 mRNA expression was strongly correlated with poor PFS in these studies [11,12,13]. We postulated that PD-L1 expression may be primarily intracellular and could be a contributing negative factor for OS or PFS in patients with ovarian cancer. Emerging evidence suggests a critical role of tumor-cell-intrinsic PD-L1 in tumor progression and the responses to chemotherapy and immunotherapy [14,15,16]. Depletion of PD-L1 has been shown to inhibit tumor proliferation in several studies [17,18]. The acetylation and deacetylation of PD-L1 appear to play important roles in the nuclear translocation of PD-L1, in a breast cancer model, to regulate the efficacy of immunotherapy [15]. PD-L1 is also required for DNA double-strand break repair in the cell nucleus [16,19]. Recent studies have demonstrated that PD-L1 regulates gene expression, including the suppression of p21 and STING expression [20,21,22]. The STING pathway can also be activated after sensing DNA double-strand breaks, leading to G2/M arrest through the upregulation of p21 [23] and cell apoptosis [24]. Importantly, a recent study suggested that genetic PD-L1 deficiency, but not cell-surface PD-L1 inhibition by antibodies, sensitizes tumors to PARPis in vivo [16]. Additionally, intracellular PD-L1 can reduce the sensitivity of tumor cells to PARPis [16]. However, it remains unknown whether PARPi treatment can upregulate intracellular PD-L1 to promote therapy resistance and/or cancer progression in ovarian cancer.

In this study, we show that PD-L1 expression in ovarian cancer-patient tumors is predominantly intracellular. Strikingly, PARPi treatment highly induced intracellular PD-L1 accumulation in both ovarian cancer-patient tumor samples and cell lines in vitro, prompting us to investigate the role of intracellular PD-L1 in promoting ovarian cancer progression.

## 2. Materials and Methods

### 2.1. Cell Lines and Cell Culture

The epithelial ovarian cancer-cell lines PEO1 and ID8 were purchased from Sigma, St. Louis, MO, USA. The SKOV3 cell line was kindly provided by Dr. Edward Wang, City of Hope Comprehensive Cancer Center, Duarte, and the ID8*^Brca1−/−^* was a generous gift from Dr. Jean Zhao from the Dana-Farber Cancer Institute at Harvard Medical School in Boston, MA, USA. All of the cells were cultured in DMEM (Gibco, Grand Island, NY, USA) supplemented with 10% fetal bovine serum (FBS) and 1% penicillin/streptomycin at 37 °C and 5% CO_2_, with frequent mycoplasma checks with a MycoAlert^®^ detection kit (Lonza, Cambridge, MA, USA, #LT07-318). For the tumor-cell spheroid culture, OVCAR8 cells pre-stained with the Qtracker™ 655 Cell Labeling Kit (Invitrogen, Waltham, MA, USA, #Q25021MP) were seeded into 96-well ultralow attachment (ULA) plates at 1 × 10^3^ cells per well and cultured for 2 days prior to treatment.

### 2.2. Patients and Tumor Samples

Female ovarian cancer-patient specimens were obtained through protocols approved by the City of Hope Institutional Review Board (IRB18004 and IRB07047), as mentioned previously [25,26]. For the patient samples from normal ovarian tissue, a tumor site, and a metastatic lesion, tissue samples were obtained from one patient diagnosed with HGSOC. Paired tumor specimen sections (before and after PARPi therapy, formalin-fixed and paraffin-embedded) were obtained from 3 patients with germline BRCA mutations.

### 2.3. Acetylation Site Point Mutation and Stable Cell Line Establishment

Human (#RC213071) and mouse (#MR203953) PD-L1 plasmids were purchased from OriGene, Rockville, MD, USA, and the phosphorylated primers (human: F: 5′-P-atc ttc cgt tta aga cga ggg aga atg atg gat-3′; R: 5′-P- atc cat cat tct ccc tcg tct taa acg gaa gat-3′, mouse: F: 5′-P- ctc ctc ttc ttg aga cga caa gtg aga atg cta -3′; R: 5′-P- tag cat tct cac ttg tcg tct caa gaa gag gag-3′) were used to generate acetylation site mutation (Lys to Arg) via PCR-based site-directed mutagenesis. DpnI (NEB, Ipswich, MA, USA, # R0176S) was applied at 37 °C for 1 h to the PCR products to remove the DNA templates. Competent *E. coli* cells (NEB 5-alpha, Ipswich, MA, USA, #C2987H) were transformed with the PCR products. Kanamycin-resistant individual colonies were grown overnight at 37 °C in LB broth containing kanamycin (25 µg/mL). DNA samples were extracted and sent for sequencing (Sanger sequencing), and then selected for the desired mutation. After that, plasmid DNA was prepared with a QIAGEN Plasmid Maxi Kit (Qiagen, Hilden, Germany, #12163) according to the manufacturer’s manual. Wild-type (WT) and mutant (Mut) DNA were then transfected into PEO1 or ID8*^Brca1−/−^* cells with Lipofectamine™ LTX Reagent with PLUS™ Reagent (Invitrogen, #15338100) according to the manufacturer’s manual. Single clones were collected and cultured in selection media supplemented with 800 µg/mL G418 (Geneticin, Gibco, Grand Island, NY, USA, #10131035) to establish stable cell lines that overexpress WT or Mut PD-L1.

### 2.4. Cell-Penetrating Antibody Conjugation

Cell-penetrating antibody conjugation was carried out via similar approaches, as described previously [27]. Briefly, a fully phosphorothioated (PS) single-stranded DNA (ssDNA) 20-mer was synthesized with a 5′ extension featuring C3 (propyl) spacers (iSpC3) and an azide reactive group for antibody conjugation. The antibodies were modified with DBCO-TEG prior to processing with PS-conjugation click chemistry. Free PS-oligo was removed by a size exclusion column, and the PS-antibody was dialyzed in 1× HBSS buffer before use. Atezolizumab biosimilar (SIM0009) and its isotype control (human IgG1, CP171), as well as rat anti-mouse PDL1 (10F.9G2, BE0101) and rat IgG2b isotype control (BE0090), were sourced from BioXCell, Lebanon, NH, USA Atezolizumab was purchased from Genentech, South San Francisco, CA, USA.

### 2.5. Western Blotting

Total cellular protein was extracted via radioimmunoprecipitation assay (RIPA) buffer supplemented with Halt™ protease and phosphatase inhibitor cocktail (Thermo Fisher Scientific, Waltham, MA, USA, #1861281). Protein concentrations were determined by BCA, using a Thermo Fisher Scientific kit (#23225). Normalized protein amounts were subjected to sodium dodecyl sulfate-polyacrylamide gel electrophoresis (SDS–PAGE; 10% polyacrylamide gels) or NuPAGE™ Bis-Tris Mini Protein Gels (Invitrogen, #NP0322BOX); then, they were transferred to TransBlot^®^ Turbo™ PVDF membranes (Bio-Rad, Hercules, CA, USA) for Western blotting. For cell fractioning, samples were prepared with a Cell Fractionation Kit (Cell Signaling Technology, Danvers, MA, USA, 9038). The primary antibody against β-actin (#A2228) was purchased from Sigma–Aldrich. Antibodies against GAPDH (#SC-3233) and p21 (#SC-6246) were purchased from Santa Cruz Biotechnology, Dallas, TX, USA. The antibody against cleaved caspase-3 (#9654S) was purchased from Cell Signaling Technology. The antibodies against ubiquitin (Ubi-1, # ab72S4), mouse PD-L1 (#ab213480), and gammaH2AX (γH2AX, #ab26350) were obtained from Abcam, Cambridge, UK. Anti-human PD-L1 antibodies were purchased from Cell Signaling Technology (#1368S) and Abcam (#ab236238). The antibody against phospho-STING (#PA5-105674) was purchased from Invitrogen. The blot images were developed by using a ChemiDoc™ MP Imaging system (Bio-Rad).

For immunoprecipitation, we incubated an anti-PD-L1 (Cell Signaling Technology, #1368S) antibody with precleared cell lysates overnight and pulled down the antibody–antigen complex using recombinant protein G agarose beads from Thermo Fisher Scientific. We then subjected the samples to SDS–PAGE followed by immunoblotting. Quantification of Western blot images was performed via Image Lab (Version 6.0.1) software.

### 2.6. Real-Time PCR Analysis

Total mRNA was extracted from cells using an RNeasy Plus mini kit (QiaGene, Hilden, Germany, # 74134) according to the manufacturer’s protocol. CDNA was mixed with the iScript™ Reverse Transcription Supermix for RT-qPCR (Bio-Rad, #1708840) according to the manufacturer’s protocol and analyzed with specific primers via real-time PCR with the CFX96 Real-Time PCR Detection System (Bio-Rad) and Luna University qPCR Master Mix (New England Biolabs, Ipswich, MA, USA, #M3003S). The primers for VSIR (#HP214399), BIRC3 (#HP205102), RELB (#HP209439), and PDCD1LG2 (#HP215316) were purchased from OriGene (Rockville, MD, USA). The primers for human PD-L1 (F: 5′-TGCCGACTACAAGCGAATTACTG-3′; R: 5′-CTGCTTGTCCAGATGACTTCGG-3′) and mouse PD-L1 (F: 5′-CCACGGAAATTCTCTGGTTG-3′; R: 5′-TGCTGCATAATCAGCTACGG-3′) were ordered from Integrated DNA Technologies, Coralville, IA, USA. The gene expression levels were normalized to those of the ACTB (F: 5′-AGGCACCAGGGCGTGAT-3′; R: 5′-GCCCACATAGGAATCCTTCTGAC-3′) housekeeping gene, which served as an internal control. Real-time PCR was performed in triplicate, and the relative fold change was measured via the Δ(CT) or ΔΔ(CT) method.

### 2.7. Gene Expression Analysis of Ovarian Cancer TCGA

The RNAseq data were downloaded from the TCGA PanCancer Atlas (https://www.cbioportal.org/, accessed on 14 July 2022). The correlations between the expression patterns of genes of interest and the PD-L1 (CD274) in ovarian cancer-patient tumor tissues were analyzed using Spearman correlation via GraphPad Prism.

### 2.8. Cell Viability Assays

The cells were plated on a 96-well plate at 1 × 10^3^ cells per well and treated as indicated in the figure legends, and the cell viability was analyzed via the CellTiter-Glo^®^ Luminescent Cell Viability Assay (Promega, Madison, WI, USA, #G7570), following the manufacturer’s instructions. Luminescence was recorded via a Cytation 5 Cell Imaging Multi-Mode Reader (BioTek, Winooski, VT, USA).

### 2.9. Colony Formation Assay

The cells were seeded into 6-well plates at 500–3000 cells/well, cultured for 7–14 days, fixed with ice-cold methanol, stained with crystal violet, washed with water, and photographed via a ChemiDoc MP Imaging System (Bio-Rad).

### 2.10. MG-132 Assay

The cells were plated and treated with 25 μM MG-132 (Sigma Aldrich, St. Louis, MO, USA, #M7449), and then harvested at the indicated time points. Protein was extracted from the cells and subjected to Western blot analysis according to the protocol listed above.

### 2.11. Cell Cycle Analysis

The cells were trypsinized, transferred to 5 mL BD Falcon tubes (BD Biosciences, Franklin Lakes, NJ, USA), and then mixed with cold 70% ethanol on ice. After fixation, the cells were washed and incubated with propidium iodide (PI) and RNase A for flow cytometry analysis. An Attune NxT flow cytometer (Thermo Fisher Scientific) and Invitrogen Attune Cytometric Software (Version 5.3.2) were used for instrument control and data acquisition. Data analysis was performed via FlowJo 10.8.1.

### 2.12. Apoptosis

The cells were collected and washed with ice-cold PBS twice, after which 100 µL of the following incubation reagent was added to each sample: 10 µL of 10× Annexin V binding buffer (BD Pharmingen, BD Biosciences #51-66121E), 1 µL of propidium iodide (PI), 5 µL of Annexin V (BioLegend, San Diego, CA, USA, #640943), and 84 µL of distilled water. Four hundred microliters of 1× binding buffer were then added to each sample after 15 min of incubation in the dark. Apoptotic cells were detected with an Attune NxT flow cytometer (Thermo Fisher Scientific). Invitrogen Attune Cytometric Software was used for instrument control and data acquisition. Data analysis was performed via FlowJo 10.8.1.

### 2.13. Immunofluorescent (IF) and Immunohistochemical (IHC) Staining

Formalin-fixed paraffin-embedded (FFPE) ovarian tumor sections were deparaffinized at 65 °C for 1 h and then rehydrated through the use of xylene and an ethanol series; this was followed by antigen retrieval in sodium citrate buffer (Vector, Newark, CA, USA, #H-3300), pH 6.0, and signal enhancement via image-iT™ Fx-Signal Enhancer (Invitrogen, #I36933). Slides were then blocked with background-reducing components (Dako, Glostrup, Denmark, #S3022); this was followed by incubation with primary antibodies (PD-L1 (Cell Signaling Technology, #86744S), Ki67 (Vector, Newark, CA, USA, #VP-RM04), pancytokeratin (Novus, Chesterfield, MO, USA, #NBP2-33200AF647), E-cadherin (Cell Signaling Technology, #3195S), P21 (Santa Cruz Biotechnology, Dallas, TX, USA, #SC-6246), or γH2AX (Abcam, #ab26350)) at 4 °C overnight. The samples were then incubated with secondary antibodies using Biocare Medical Van Gogh Diluent (#PD902H) with either MACH 2 mouse (#MHRP520G) or MACH 2 rabbit (#RHRP520G) antibodies. Fire-red Opal reagent was then added for color (AKOYA, Marlborough, MA, USA, #OP-001003). In the end, the slides were counterstained with Hoechst 33342 or DAPI. Images were acquired via a Zeiss (Oberkochen, Germany) LSM700, LSM880, or LSM900 confocal microscope. Staining quantification was performed with ZEN 2.3 lite software, and the results were plotted with GraphPad Prism 8 software. For Ki67 analysis, nuclear positive percentage within tumor area was calculated by using the software QuPath-0.4.2.

Frozen ovarian tumor sections were briefly dried under a chemical hood and then fixed in 2% paraformaldehyde for 10 min and then permeabilized with −20 °C methanol. Slide enhancement, blocking, primary antibody incubation, secondary antibody incubation, Opal staining, counterstaining, and mounting were then performed in a manner identical to that of the FFPE slide staining.

IHC staining in our lab was performed with a Pierce™ Peroxidase IHC Detection Kit according to the manufacturer’s instructions.

For the IHC staining for clinical pathology review, unstained sections at 4 µm thickness were obtained from formalin-fixed paraffin-embedded (FFPE) tumor tissue blocks. SP263 assay (Ventana Medical Systems Inc., Tucson, AZ, USA) by Ventana BenchMark Ultra was used, and the staining was performed according to the manufacturer’s instructions.

### 2.14. In Vivo Experiments in Xenograft Tumor Models

Animal care and experiments were conducted following approved institutional IACUC protocols (10003 and 08026) from the Research Animal Care Committees of the City of Hope. To create a WT/Mut PD-L1 tumor-bearing mouse model, we subcutaneously injected 5 × 10^6^ PEO1- or ID8*^Brca1−/−^* overexpressing WT or Mut PD-L1 cells into eight- to twelve-week-old female NOD Scid Gamma (NSG) mice. Tumor growth was measured with a digital caliper 2–3 times per week.

For the treatment experiment, 5 × 10^6^ ID8*^Brca1−/−^* overexpressing WT PD-L1 cells were subcutaneously injected into female C57BL/6 mice. When the tumor size reached 100 mm^3^, the tumor-bearing mice were randomly divided into five groups. Each group (*n* = 5) of mice was treated with PBS, IgG, an α-PD-L1 (anti-PD-L1) antibody (atezolizumab), PS-IgG, or PS-α-PD-L1 at 100 μg/mouse 3 times per week via i.p. injections for a total of 6 treatments. Tumor growth was measured with a digital caliper 3 times per week.

The mice were monitored daily for any signs of pain or distress. The maximum tumor size permitted was 1500 mm^3^, and the tumor sizes did not exceed this limit in the present study.

### 2.15. Statistical Analysis

Student’s *t*-test (two-sided) was used for comparisons between groups. For comparisons of more than 2 groups, one-way ANOVA was used. Values of *p* < 0.05 were considered statistically significant. In the figures, changes are noted using * *p* < 0.05, ** *p* < 0.01, and *** *p* < 0.001. Statistical analysis was conducted via the Microsoft Excel or GraphPad Prism software (Version 9.3.0) packages.

## 3. Results

### 3.1. PD-L1 Is Largely Intracellular in Ovarian Cancer Cells, Especially Following PARPi Treatment

Using immunofluorescence staining, we compared tissue sections from normal ovarian and primary tumor and metastatic tissues from one patient. We observed an increase in intracellular PD-L1 expression in the malignant tissues over the normal tissue (Figure 1A), while there was no difference in PD-L1 expression between the primary tumor and metastatic tissues. We also assessed PD-L1 expression levels in tumor tissues from three ovarian cancer patients before and after PARPi treatment and detected a significant increase in the intracellular accumulation of PD-L1, with minimal to no increase in cell membrane expression (Figure 1B). Immunohistochemistry staining further supported the results from immunofluorescent images, confirming that the level of intracellular PD-L1 increased after PARPi treatment (Appendix A). Notably, samples with high intracellular PD-L1 expression by IHC were scored as “0/no expression” by scoring methods used in the clinical setting, which only score surface membranous expression of PD-L1. To further validate our results, we performed immunohistochemistry (IHC) staining of the three patient-matched pairs of samples with a commercially available Ventana PD-L1 SP263 antibody kit and the protocol used in the routine clinical setting for PD-L1 expression evaluation, in a CLIA certified laboratory. The IHC slides were reviewed by a gynecologic pathology clinical faculty member involved in this study, and similar staining results were provided. For in vitro study, we treated mouse ovarian cancer cells ID8*^Brca1−/−^* and human ovarian cancer cells PEO1 and OVCAR8 cells with PARPi olaparib in vitro. We found markedly increased intracellular PD-L1 accumulation (Figure 1C and Appendix A). Furthermore, PD-L1 expression was significantly increased in the olaparib-resistant PEO1 cells relative to that of the parental cells (Appendix A). Microscopic imaging further showed that the increase in PD-L1 was largely intracellular (Appendix A), suggesting a potential role of intracellular PD-L1 in PARPi drug resistance in ovarian cancer.

### 3.2. Acetylation Site Mutation Decreases the Expression of PD-L1 in Ovarian Cancer Cells

Acetylation at the lysine sites of nonhistone proteins was shown to compete with ubiquitination, altering protein stability or subcellular localization [28,29]. Post-transcriptional modifications of PD-L1, including acetylation and ligase-mediated ubiquitination degradation, have been reported in several studies [30,31,32,33,34]. Recent studies have shown that the acetylation of lysine sites in PD-L1 is crucial for its transcriptional function [15]. Therefore, we investigated whether mutating the lysine acetylation site of PD-L1 affects ovarian cancer-cell growth arrest or apoptosis. We substituted the lysine site (K) (K263 in human and K262 in mouse cells) with arginine (K to R mutation) in a PD-L1 expression vector. We then performed DNA transfection to generate PEO1 and ID8*^Brca1−/−^* cell lines stably expressing WT or Mut PD-L1. Stable cell lines with similar mRNA levels of PD-L1 were selected for the study (Figure 2A). We found that total PD-L1 protein levels were significantly lower in the Mut cells than in the WT cells (Figure 2B). Further subcellular fractionation analysis revealed that the PD-L1 was decreased, both intracellularly and on the cell membrane, in the Mut cells (Figure 2C). To test whether degradation was involved in PD-L1 downregulation due to mutation of the acetylation site, we treated ovarian cancer cells with the proteasome inhibitor MG-132. The data revealed that MG-132 treatment reversed the decrease in PD-L1 expression induced by mutation of the acetylation site (Appendix A). The results of co-immunoprecipitation Western blotting suggest that PD-L1 degradation can be attributed to its ubiquitination (Appendix A).

### 3.3. Decreasing PD-L1 Inhibits Ovarian Cancer-Cell Colony Formation and Induces Cell Cycle Arrest and Apoptosis

Next, we investigated whether decreasing PD-L1 via acetylation site mutation could inhibit ovarian cancer-cell growth. Using a colony formation assay, we showed that a lack of intracellular PD-L1 in PEO1 and ID8*^Brca1−/−^* cells reduced tumor-cell proliferation (Figure 3A). Proliferation inhibition in the Mut PD-L1-expressing ovarian cancer cells was accompanied by increased G2/M cell cycle arrest (Figure 3B) and apoptosis induction (Figure 3C,D). To elucidate the potential mechanism underlying decreased cell cycle arrest and increased apoptosis induced by reduced intracellular PD-L1, we assessed the expression levels of cleaved caspase-3 and p21, which are known to be suppressed by PD-L1 [21,35]. Western blot analysis revealed that the level of the proapoptotic protein cleaved caspase-3 was increased in the Mut cells (Figure 3E). Several studies have shown a role for p21 in G2/M phase arrest [36,37,38]. Our western blot results revealed an increase in p21 in the PD-L1 Mut cells (Figure 3E). These results suggest that targeting intracellular PD-L1 can inhibit ovarian cancer-cell growth through G2/M arrest and apoptosis induction.

### 3.4. Decreasing Intracellular PD-L1 Expression via Acetylation Site Mutation Inhibits Ovarian Tumor Growth In Vivo

We next investigated the effect of the downregulation of PD-L1 expression through acetylation site mutation on tumor growth in vivo. We subcutaneously injected PEO1 (Figure 3F) or ID8*^Brca1−/−^* (Figure 3H) ovarian tumor cells into NSG mice. Our results revealed that the growth of the Mut tumors was significantly slower than that of the WT tumors. Tumor tissues were harvested at the endpoint, and proliferation rates were assessed via Ki67 staining. The results indicated that there was a significant reduction in Ki67+ cells in the tumor sections from the Mut group in both animal models (Figure 3G,I). These results further indicate that PD-L1 contributes to ovarian cancer-cell proliferation and tumor growth.

### 3.5. Downregulation of PD-L1 Enhances DNA Damage and STING Activation in Ovarian Cancer Cells

Next, we investigated the possible mechanisms through which intracellular PD-L1 regulates tumor progression. Prior studies have suggested that blocking tumor-cell-intrinsic PD-L1 can cause DNA damage, as marked by an increase in γH2AX [16,19], and that STING can be subsequently activated after DNA damage, leading to cycle arrest through increased p21 [20] and apoptosis via caspase-3 cleavage [22] in other tumor cells. Our current study showed that both p21 and cleaved caspase-3 increased when PD-L1 was decreased by mutating the acetylation site (Figure 3E). We therefore tested the effects of PD-L1 acetylation mutation on the level of γH2AX and the phosphorylation of STING. Our data indicated that a decrease in PD-L1 expression caused by an acetylation site mutation increased levels of γH2AX and activated STING (pSTING) (Figure 3E). These results suggest that the downregulation of PD-L1 through acetylation mutation in ovarian cancer cells can cause DNA damage, which in turn can activate STING and its downstream genes, such as p21 and caspase-3, to mediate cell cycle arrest and apoptosis.

### 3.6. Targeting Intracellular PD-L1 with a Cell-Penetrating Antibody Decreases Ovarian Cancer-Cell Proliferation In Vitro

Although mutating or deleting the PD-L1 gene can provide important information on the role of intracellular PD-L1 in ovarian cancer progression [15,16], it is necessary to employ approaches to inhibit intracellular PD-L1 in a manner that is both physiologically and clinically relevant. We have recently developed a technology that enables antibodies, peptides, and CRISPR to penetrate cells and nuclei by attaching phosphorothioate DNA oligos (PS) [27,39,40]. For this study, we generated cell-penetrating PD-L1 antibodies (PS-α-PD-L1) by covalently attaching PS to anti-human PD-L1 antibodies (atezolizumab) and anti-mouse PD-L1 antibodies. Next, we targeted intracellular PD-L1 in ovarian cancer cells with either anti-human or anti-mouse PS-α-PD-L1. To show the cell-penetrating efficiency in ovarian cancer cells, we used fluorescein-labeled PS-α-PD-L1 and PS-IgG (isotype control). The fluorescence microscopy images revealed that PS-α-PD-L1 was efficiently internalized in PEO1 ovarian cancer cells after 6 h of incubation (Figure 4A). PS-IgG-treated PEO1 cells did not show a strong fluorescent signal, which could be explained by the earlier finding that stable intracellular retention of the PS-antibody requires the presence of the antibody target antigen [27]. We also showed that PS-α-PD-L1 penetrated 3-D tumor spheroids derived from OVCAR8 ovarian cancer cells after 24 h of incubation (Appendix A). Furthermore, the intracellular PD-L1 level was significantly decreased after PS-α-PD-L1 treatment (Figure 4B and Appendix A).

Next, we determined the effects of PS-α-PD-L1 on tumor-cell viability. Compared to those treated with unmodified α-PD-L1 (atezolizumab), PBS, IgG, or PS-IgG, PS-α-PD-L1-treated PEO1 cells presented a reduced viability of approximately 30% (Figure 4C). Western blot analysis also revealed that the PD-L1 protein level was lower in the PS-α-PD-L1 antibody-treated group than in the control groups (Figure 4D and Appendix A).

Based on the results obtained by mutating the PD-L1 acetylation site (Figure 3E), we evaluated PS-α-PD-L1 treatment on the expression of p21, γH2AX, pSTING, and cleaved-caspase 3. The Western blot results showed that they were all upregulated after PS-α-PD-L1 antibody treatment (Figure 4D). An annexin V staining assay also revealed that, compared to the other treatments, the PS-α-PD-L1 antibody induced significantly more apoptosis, especially early apoptosis, in the tumor cells (Figure 4E). Taken together, these results are consistent with those observed from acetylation site mutation (Figure 3), confirming that targeting intracellular PD-L1 inhibits ovarian cancer-cell proliferation and induces cell apoptosis, which could be attributed to DNA damage induction, as indicated by the increase in γH2AX (Figure 4F).

### 3.7. Cell-Penetrating PS-α-PD-L1 Antibody Inhibits Tumor Growth In Vivo

Based on the results obtained by PS-α-PD-L1 antibody treatments on ovarian tumor-cell growth inhibition in vitro, we investigated the effects of targeting intracellular PD-L1 with a cell-penetrating antibody on ovarian tumor growth in vivo. Mice with established subcutaneously engrafted ID8*^Brca1−/−^* ovarian tumors overexpressing PD-L1 were treated with PBS, IgG, an α-PD-L1 antibody, PS-IgG, or PS-α-PD-L1. While the α-PD-L1 antibody and other control treatments had no significant effects on tumor growth, PS-α-PD-L1 treatment resulted in significant tumor growth inhibition (Figure 5A).

We further analyzed the tumors from all the treatment groups for tumor-cell proliferation via Ki67 immunostaining. Tumors from the mice treated with PS-α-PD-L1 presented significantly fewer proliferative cells, as indicated by the reduced number of Ki67+ cells in the tumor areas (Figure 5B). We also examined the PD-L1 expression level in tumor cells from mice treated with various antibodies. The results from immunostaining and confocal microscopic analysis revealed that PD-L1 levels in the tumors from mice treated with PS-α-PD-L1 were significantly lower than those from the mice receiving other treatments (Figure 5C,D). Furthermore, immunostaining and microscopic analyses revealed increased levels of p21 and the DNA damage marker γH2AX in tumors from mice receiving PS-α-PD-L1 treatment (Figure 5C). These in vivo results further suggest that decreasing intracellular PD-L1 protein levels via the PS-α-PD-L1 antibody induces tumor growth inhibition, which can be attributed to increases in tumor-cell DNA damage and p21.

## 4. Discussion

One of the most effective treatments for multiple solid tumors is targeting the interactions between PD-L1 on the tumor-cell surface and PD-1 on T cells. However, for ovarian cancer, the response to PD-1/PD-L1 inhibition is low. Our current study revealed that PD-L1 expression in ovarian cancer-patient tumors is mainly intracellular. Because the current clinical standards for scoring PD-L1 expression are limited to the cell membrane expression levels, ovarian cancer-patient tumors are largely considered “PD-L1 low”. We detected significant accumulation of intracellular PD-L1 in ovarian cancer-patient tumor samples post treatments including PARP inhibitor(s). Additionally, we employed FDA-approved IHC protocols used in routine clinical settings to validate our results. Our findings suggest that the low response rate to PD-L1/PD-1 blockade in ovarian cancer, especially after PARPi treatment, is contributed by the lack of tumor-cell-surface PD-L1 expression.

Among the paired patient tumor sections analyzed, there were tumors treated with olaparib, niraparib, or both. Decreasing PARP1 genetically induces pSTAT3 [41,42], and studies have shown that pSTAT3 can upregulate PD-L1 [41,43,44]. Olaparib has been demonstrated to increase pSTAT3 levels through dePARylation [41]. However, niraparib can also inhibit pSTAT3 [42]. One explanation for niraparib-induced pSTAT3 inhibition is that niraparib can also inhibit pSrc, which plays a role in increasing pSTAT3 [42]. On the other hand, as a PARP inhibitor, niraparib induces dePARylation, which is expected to increase pSTAT3. In tumors from patients who had disease progression while on niraparib therapy, pSTAT3 expression is high [25], and the level of intracellular PD-L1 is also elevated, as shown in this study (Figure 1B). Because of the complexity of niraparib in terms of regulating pSTAT3, which could impact the expression of PD-L1, especially in a cell culture with short-term treatment, we used olaparib for our in vitro experiments. We demonstrated that olaparib can potently induce intracellular PD-L1 accumulation while not increasing PD-L1 cell membrane expression. Understanding the mechanisms that could account for intracellular PD-L1 accumulation is critical. However, our current study did not explore such mechanisms. Instead, we focused on determining whether intracellular PD-L1 in ovarian cancer cells/tumors can promote tumor progression and whether targeting intracellular PD-L1 would induce ovarian cancer-cell growth inhibition and apoptosis.

In mutating the PD-L1 acetylation site, we found that the PD-L1 protein level decreased both on the cell surface and intracellularly in PEO1 and ID8*^Brca1−/−^* cells, and decreased colony formation, which was accompanied by cell cycle G2/M arrest and apoptosis induction. The growth of PEO1 or ID8*^Brca1−/−^* tumors with PD-L1 acetylation site mutations was also inhibited in vivo in NSG mice. This finding is consistent with several pioneering studies showing the role of intrinsic PD-L1 in tumor growth and DNA damage repair [16,21,45]. The acetylation and deacetylation of PD-L1 appear to play important roles in transporting PD-L1 into the nucleus in breast cancer [15]. However, in our ovarian cancer models, we have found that PD-L1 with the acetylation site mutation undergoes protein degradation, possibly mediated by ubiquitination. Consistent with our finding, it has been shown that lysine acetylation can compete with ubiquitination [28,29], and ubiquitin-mediated PD-L1 degradation has been reported [30,31,32,33,34]. Although this difference could be due to distinct cancer types or different experimental conditions, the discrepancy between PD-L1 degradation and nuclear translocation resulting from acetylation site mutation requires further investigation.

Our study, consistently with several other reports, revealed that current PD-L1 antibodies in the clinic cannot reduce the level of intracellular PD-L1 to induce antitumor effects [16,45]. Although small-molecule drugs can enter tumor cells/nuclei, to date, small-molecule PD-L1 inhibitors have been designed to interfere with PD-L1 and PD-1 cell-surface interactions. We show that targeting intracellular PD-L1 with cell-penetrating antibodies against either mouse or human PD-L1 effectively reduces ovarian tumor PD-L1, leading to tumor-cell growth inhibition and increased apoptosis both in vitro and in vivo. Prior studies have also demonstrated potent antitumor effects via the use of a unique PD-L1 antibody capable of targeting intracellular PD-L1 [46]. Our results support their findings. This unique PD-L1 antibody has the potential to be developed for future clinical use. Furthermore, modifying current FDA-approved PD-L1 antibodies to target intracellular PD-L1, as shown in our study, may also have clinical significance.

Targeting intracellular PD-L1 has been shown to promote antitumor immune responses in some mouse tumor models [15,47,48], and nuclear PD-L1 upregulates several genes involved in immunosuppression, including PDCD1LG2, BIRC3, RELB, and VSIR [15,49,50]. We also found a significant positive correlation between PD-L1 (CD274) and the PDCD1LG2, BIRC3, RELB, and VSIR genes in the TCGA ovarian cancer-patient database (Appendix A). In support of an antitumor tumor immune response, by targeting intracellular PD-L1 with a cell-penetrating antibody, the mRNA expression of PDCD1LG2, BIRC3, RELB, and VSIR was significantly decreased in ovarian cancer cells (Appendix A). In our in vivo study of ID8*^Brca1−/−^* tumors overexpressing PD-L1 in a mouse syngeneic model, PS-α-PD-L1 treatment significantly inhibited tumor growth, compared with α-PD-L1 and other control treatments. The tumor growth inhibition was also accompanied by decreased tumor proliferation and increased DNA damage. However, we did not find a significant difference in tumor immune cell infiltration, the activities of tumor-infiltrating CD8+ and CD4+ T cells, or the number of tumor-infiltrating M1/M2 TAMs between the PS-α-PD-L1 group and the other treatment groups. More studies using different ovarian tumor models are needed to assess the antitumor immune responses resulting from the targeting of tumor intracellular PD-L1. Nevertheless, our in vivo data using PS-α-PD-L1 underscore the critical tumor-cell-intrinsic role of intracellular PD-L1 in promoting ovarian cancer progression.

## Figures and Tables

**Figure 1 cells-14-00314-f001:**
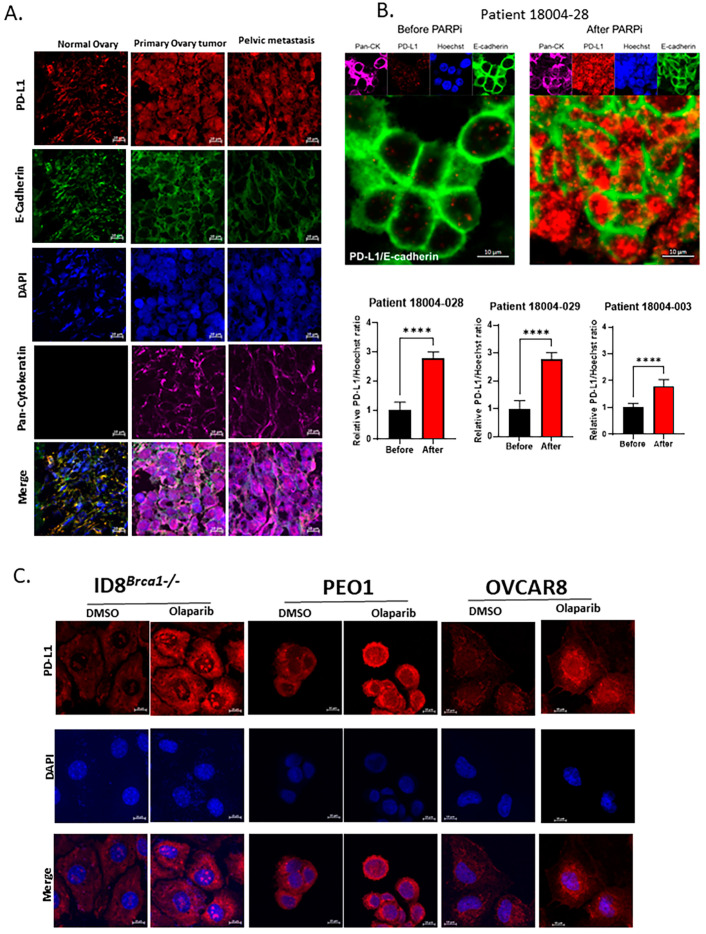
Intracellular PD-L1 accumulation in ovarian cancer-patient tumor samples and cell lines after PARPi treatment. (**A**) Immunofluorescence staining of normal ovary, primary tumor, and metastatic tissues from an ovarian cancer patient. PD-L1 staining is shown in red; pan-cytokeratin is shown in purple as a tumor marker, E-cadherin is shown in green as a cellular membrane marker, and nuclear DAPI staining is shown in blue. Scale bar, 10 μm. (**B**) Representative images of immunofluorescence staining of ovarian cancer-patient tumor samples before and after PARPi treatment. PD-L1 staining is shown in red; pan-cytokeratin is shown in purple as a tumor marker, E-cadherin is shown in green as a cellular membrane marker, and nuclear Hoechst staining is shown in blue. Scale bar, 10 μm. The bar graphs show the results of the statistical analysis for the quantification of PD-L1 levels in 3 pairs of ovarian cancer-patient tumor samples before and after PARPi treatment. Data are shown as the means ± SEMs, and Student’s *t*-test was used for statistical analysis. (**** *p* < 0.0001). (**C**) Immunofluorescence staining of PD-L1 in human and mouse ovarian cancer-cell lines PEO1, ID8*^Brca1−/−^*, and OVCAR8 following treatment with or without olaparib. Cells were cultured in growth media with 0.1% DMSO or 20 µM olaparib for 48 h. Red, PD-L1; blue, nucleus. Scale bar, 10 µm. All three cell lines presented significantly greater amounts of cytoplasmic and nuclear PD-L1 staining following olaparib treatment, compared to DMSO control treatment.

**Figure 2 cells-14-00314-f002:**
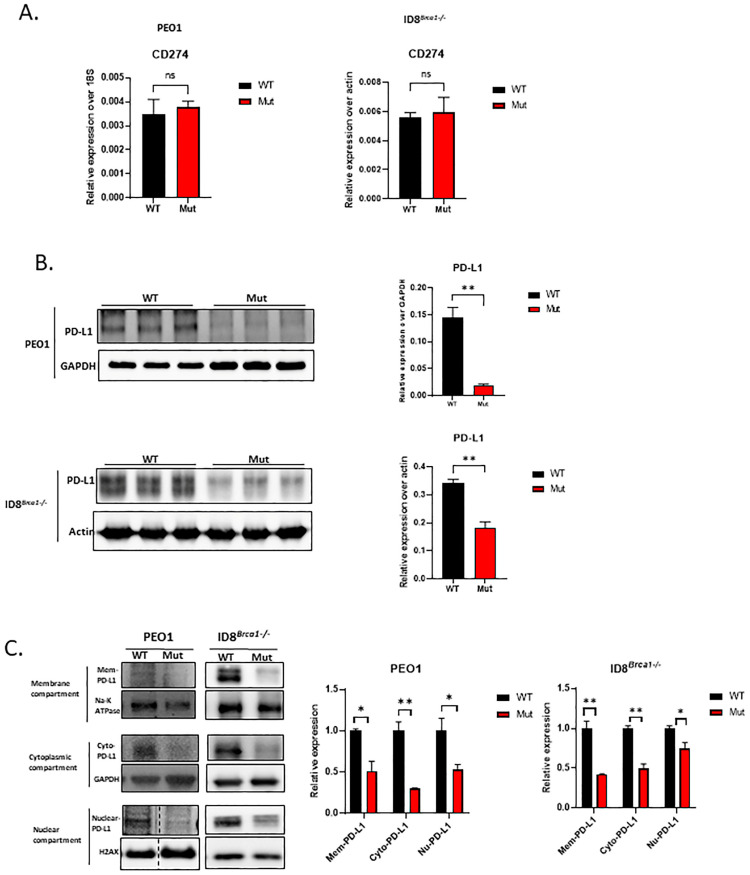
Acetylation site mutation decreases intracellular and surface levels of PD-L1 and induces PD-L1 protein degradation. (**A**) RNA levels of CD274 (PD-L1 gene) in PEO1 and ID8*^Brca1−/−^* cells overexpressing WT or Mut PD-L1. (ns= non-significant) (**B**) Western blot analysis of PD-L1 in PEO1 and ID8*^Brca1−/−^* cells overexpressing WT or Mutant PD-L1. The bar graphs show the results of the statistical analysis of triplicate lysate samples from 3 independent experiments. The data are shown as the means ± SEMs, and Student’s *t*-test was used for statistical analysis. (** *p* < 0.01). (**C**) Western blot analysis of membrane, cytoplasmic, and nuclear fractions derived from PEO1 and ID8*^Brca1−/−^* PD-L1 WT or Mut cells. Na-K ATPase, GAPDH, and H2AX were loading controls for each compartment. The bar graphs show the results of the statistical analysis of triplicate lysate samples from 3 independent experiments. The data are shown as means ± SEMs, and Student’s *t*-test was used for statistical analysis. (* *p* < 0.05, ** *p* < 0.01).

**Figure 3 cells-14-00314-f003:**
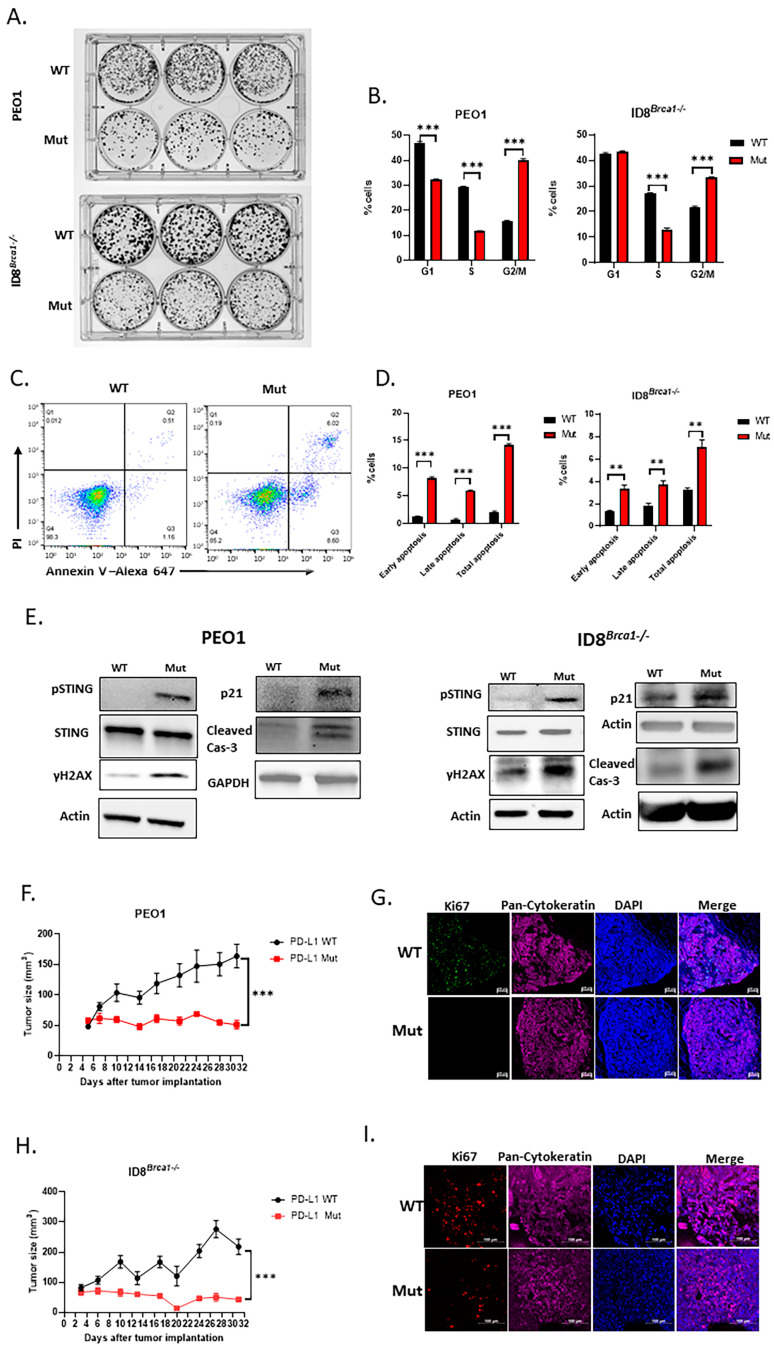
Acetylation site mutation suppresses ovarian cancer progression in vitro and inhibits tumor growth in vivo. (**A**) Representative image of colony formation assay of WT and Mut PEO1 and ID8*^Brca1−/−^*. Cells (500–3000) were seeded in triplicate in 6-well plates, incubated for 7 days, and then stained with crystal violet. (**B**) Cell cycle analysis of WT and Mut PEO1 and ID8*^Brca1−/−^* cells. The data are shown as means ± SEMs, and Student’s *t*-test was used for statistical analysis (*n* = 3, *** *p* < 0.001). (**C**) Representative flow cytometry analysis of annexin-V and PI staining as indicators of apoptosis in WT and Mut PEO1 cells. (**D**) Flow cytometry analysis of cell counts of early apoptotic (PI-, Annexin-V+) and late apoptotic (PI+, Annexin-V+) cells in WT and Mut PEO1 and ID8*^Brca1−/−^*. Data are shown as means ± SEMs, and Student’s *t*-test was used for statistical analysis (*n* = 3, ** *p* < 0.01, *** *p* < 0.001). (**E**) Western blot analysis of STING, pSTING, γH2AX, p21, caspase-3 (cas-3) and cleaved-caspase3 (cleaved cas-3) in WT or Mut PEO1 cells and ID8*^Brca1−/−^* cells. GAPDH or actin was used as loading controls. (**F**,**H**) NSG mice were injected with 5 × 10^6^ WT or Mut PEO1 and ID8*^Brca1−/−^* cells in right flanks. The tumor size was recorded twice a week. Data are shown as the means ± SEMs, and two-way ANOVA was used for statistical analysis (*n* = 5, *** *p* < 0.001). (**G**,**I**) Representative images of immunofluorescence staining of Ki67 in mouse xenograft tumor tissues from (**F**,**H**), Ki67 is shown in green (**G**) or red (**I**), pan-cytokeratin is shown in purple as a tumor marker, and nuclear Hoechst staining is shown in blue. Scale bar, 20 µm.

**Figure 4 cells-14-00314-f004:**
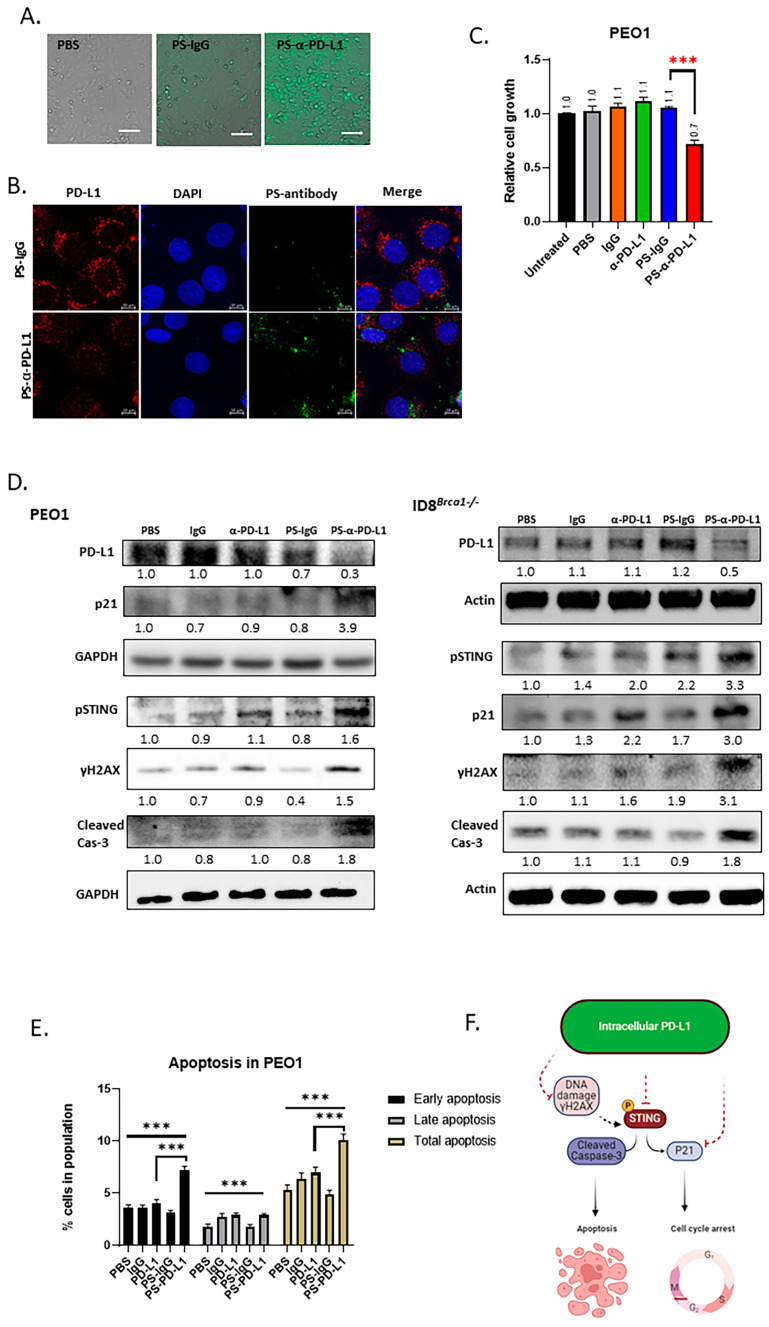
Targeting intracellular PD-L1 inhibits tumor progression and induces DNA damage and STING activation. (**A**) Microscope images showing that PS-α-PD-L1 antibody labeled with fluorescein (FITC) penetrated PEO1 cells. Scale bar, 50 µm. (**B**) PS-α-PD-L1 antibody (green) targeted PD-L1 (red color) inside tumor cells and decreased PD-L1 protein level PEO1 cells. Nuclear DAPI staining is shown in blue. Scale bar, 10 µm. (**C**) Viability analysis of PEO1 cells treated with PS-α-PD-L1 antibody. Cells were treated with PBS, IgG, α-PD-L1, PS-IgG, and PS-α-PD-L1 for 5 days, and cell viability was determined via cell titer glow assay. The data are expressed as the means ± SEM, and one-way ANOVA and Student’s *t*-test were used for statistical analysis (*n* = 3, *** *p* < 0.001). (**D**) Western blot analysis of PD-L1, pSTING, γH2AX, P21, and cleaved-cas3 after treatment with PBS, IgG, α-PD-L1, PS-IgG, or PS-α-PD-L1 for 3 days, in PEO1 and ID8*^Brca1−/−^* cells. GAPDH or actin was used as a loading control. The samples derive from the same experiment and gels/blots were processed in parallel. Quantification of target protein levels relative to loading control is labeled below the lanes. (**E**) Flow cytometry analysis of cell counts of early apoptotic (PI-, Annexin-V+) and late apoptotic (PI+, Annexin-V+) cells in PEO1 cells treated with PBS, IgG, α-PD-L1, PS-IgG, or PS-α-PD-L1 for 5 days. Data are shown as the means ± SEMs, and one-way ANOVA and Student’s *t*-test were used for statistical analysis (*n* = 3, *** *p* < 0.001). (**F**) Schematic diagram of the proposed mechanism based on the study. The solid line indicates direct effect, and the dashed line indicates indirect effect.

**Figure 5 cells-14-00314-f005:**
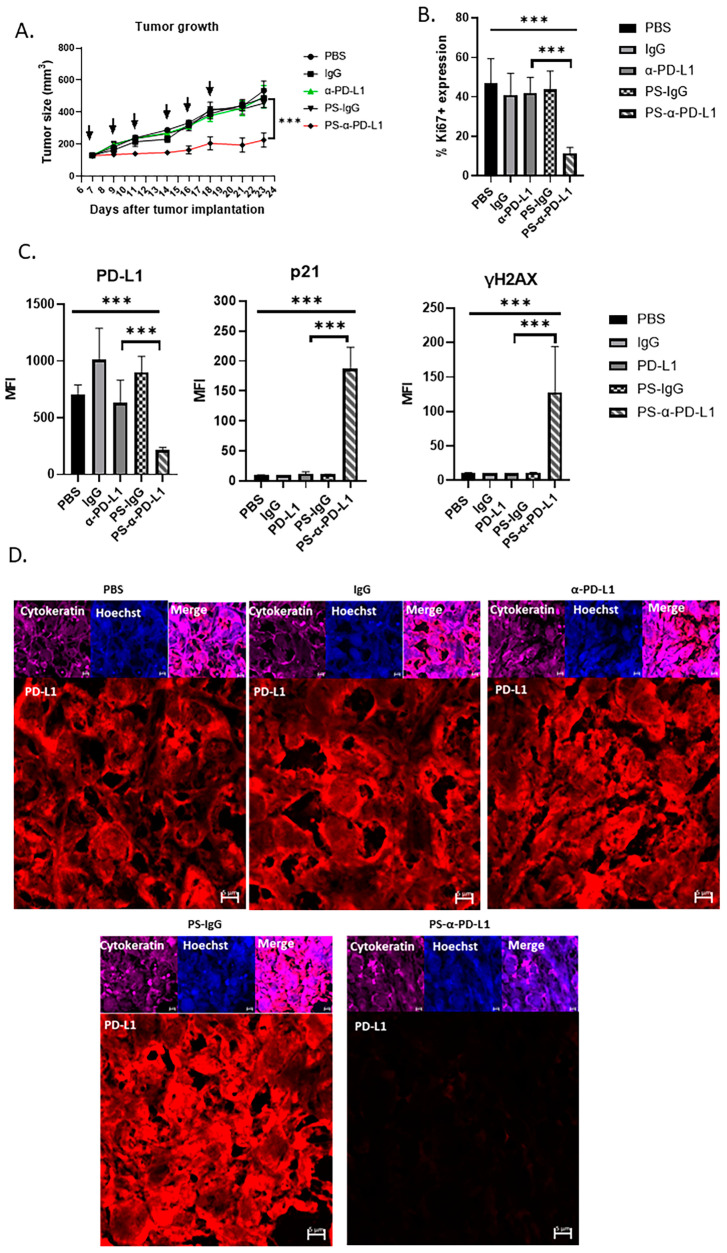
Blocking intracellular PD-L1 with PS-α-PD-L1 antibody inhibits ovarian tumor growth in vivo. (**A**) Mouse ID8*^Brca1−/−^* ovarian tumors overexpressing PD-L1 (subcutaneously) were allowed to grow until they reached approximately 100 mm^3^ prior to treatment initiation (day 7). Mice were given PBS, or 100 µg of the following antibodies: IgG, α-PD-L1, PS-IgG, or PS-α-PD-L1, three times per week for a total of six treatments (black arrow). Tumor size was measured and is presented as the means ± SEMs (*n* = 5). Two-way ANOVA was used for statistical analysis. *** *p* < 0.001. (**B**) Quantification of Ki67+ cells in tumor areas in mouse xenograft tumor tissues from (**A**). Data are shown as the means ± SD; one-way ANOVA and Student’s *t*-test were used for statistical analysis (*** *p* < 0.001). (**C**) Bar graphs show the results of the statistical analysis for the quantification of tumor PD-L1, p21, and γH2AX levels in mouse xenograft tumor tissues from (**A**). Data are shown as the means ± SDs, and one-way ANOVA and Student’s *t*-test were used for statistical analysis (*** *p* < 0.001). (**D**) Representative images of immunofluorescence staining of PD-L1 (red) in mouse xenograft tumor tissues from (**A**). Pan-cytokeratin is shown in purple as a tumor marker, and nuclear Hoechst staining is shown in blue. Scale bar, 5 µm.

## Data Availability

The data in the current study are available upon reasonable request.

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
