# Peer review of "A Critical Role of Intracellular PD-L1 in Promoting Ovarian Cancer Progression"

_cells, 2025, doi:10.3390/cells14040314_

Round 1
Reviewer 1 Report
Comments and Suggestions for Authors
The study conducted by Huang et al. presents the role of intracellular PD-L1 in ovarian cancer progression and treatment resistance, providing critical insights into the low efficacy of PD-1/PD-L1 inhibitors in this cancer type. The findings reveal that intracellular PD-L1 accumulation, particularly post-treatment with PARP inhibitors, contributes to the limited response to PD-1/PD-L1 blockade by evading detection via conventional cell surface-based scoring methods. The study’s use of FDA-approved IHC protocols lends credibility to these results, making them clinically relevant.
Comments:
The RAW data is well provided.
Was the WB membrane a PVDF or nitrocellulose
Would love to know what instrument was used to develop the blots.
CFU - Maybe if all the 4 plates were placed in a single picture it would help with data reliability.
Author Response
1. The RAW data is well provided.
Response: We appreciated the reviewer’s complement.
2. Was the WB membrane a PVDF or nitrocellulose
Response: The WB membrane was TransBlot® Turbo™ PVDF membrane, and we added this information to the Materials and Methods section.
3. Would love to know what instrument was used to develop the blots.
Response: The blot images were developed by using ChemiDoc™ MP Imaging system (Bio-Rad). We also added this information to the Materials and Methods section.
4. CFU - Maybe if all the 4 plates were placed in a single picture it would help with data reliability.
Response: We appreciate the reviewer’s suggestion. We now show triplicates of WT and Mut in one image for both cell lines in the figure.
Reviewer 2 Report
Comments and Suggestions for Authors
In the article "A critical role of intracellular PD-L1 in promoting ovarian cancer progression", the authors demonstrated the critical cell-intrinsic role of intracellular PD-L1 in promoting ovarian cancer progression. The results are novel, original and relevant as they justify the development of anti-PD-L1 immunotherapies using cell internalizing antibodies, which would greatly improve the response to therapy in cases of ovarian and many other cancers with high intracellular PD-L1 expression in cancer cells. The paper is very well written and organized. The methodology related to the in vitro (cell-penetrating antibody preparation, western blotting, immunofluorescence, immunohistochemistry, PCR, cell viability assays, etc.) and in vivo (tumor models with mutated PD-L1 expression genes, etc.) evaluation studies is well described and the results shown in all figures are convincing. The discussion covers the relevant and novel points of the research.
Minor recommendation:
In Figure 4D, it would be useful to also show the results comparing the different optical densities (relative expression) of Western blotting, as in Figure 2.
Author Response
Minor recommendation:
In Figure 4D, it would be useful to also show the results comparing the different optical densities (relative expression) of Western blotting, as in Figure 2.
Response:
We appreciate the reviewer’s positive comments and suggestions. We have now added the quantification measurement for the relative expression of target proteins in Figure 4D.